# Implications of the Organ-Specific Immune Environment for Immune Priming Effect of Radiotherapy in Metastatic Setting

**DOI:** 10.3390/biom13040689

**Published:** 2023-04-18

**Authors:** Julien Pierrard, Geneviève Van Ooteghem, Marc Van den Eynde

**Affiliations:** 1UCLouvain, Center of Molecular Imaging, Radiotherapy and Oncology (MIRO), Institute de Recherche Experimentale et Clinique (IREC), 1200 Brussels, Belgium; 2Radiation Oncology Department, Cliniques Universitaires Saint-Luc, 1200 Brussels, Belgium; 3Medical Oncology Department, Cliniques Universitaires Saint-Luc, 1200 Brussels, Belgium

**Keywords:** tumour immune microenvironment, radio-immunotherapy combination, liver metastases, liver immune system

## Abstract

With the development of immune checkpoint inhibitors (ICIs), the tumour immune microenvironment (TIME) has been increasingly considered to improve cancer management. The TIME of metastatic lesions is strongly influenced by the underlying immune contexture of the organ in which they are located. The metastatic location itself appears to be an important prognostic factor in predicting outcomes after ICI treatment in cancer patients. Patients with liver metastases are less likely to respond to ICIs than patients with metastases in other organs, likely due to variations in the metastatic TIME. Combining additional treatment modalities is an option to overcome this resistance. Radiotherapy (RT) and ICIs have been investigated together as an option to treat various metastatic cancers. RT can induce a local and systemic immune reaction, which can promote the patient’s response to ICIs. Here, we review the differential impact of the TIME according to metastatic location. We also explore how RT-induced TIME modifications could be modulated to improve outcomes of RT-ICI combinations.

## 1. Introduction

Due to recent advances in the understanding of cancer, tumoral cells are no longer considered the only target for treatment. The tumour microenvironment also plays an important role in the disease process and response to treatment. A growing body of evidence highlights the significant role of tumour-infiltrating immune cells, which are directly involved in immune destruction evasion, a new cancer hallmark [1]. Targeting immune evasion has led to one of the most significant advances in cancer treatment, namely immunotherapy. The prognosis for both locally advanced and metastatic tumours is significantly improved by inhibiting immunological checkpoints that suppress antitumoral immune responses, such as programmed cell death (PD-1), its ligand PD-L1, or cytotoxic T-lymphocyte-associated protein 4 (CTLA-4). However, the effects of immune checkpoint inhibitor (ICI) immunotherapy differ considerably across tumour histology. Excellent outcomes have been demonstrated for non-small cell lung cancer (NSCLC), melanoma, and colorectal cancer (CRC) with high microsatellite instability (MSI)/deficiency in mismatch repair (dMMR) mechanisms. However, results for other primary cancers, such as prostate cancer, hormone-sensitive breast cancer, and CRC with microsatellite stability (MSS) have been less promising [2,3,4,5,6]. One hypothesis for this response heterogeneity is that it is highly dependent on the degree of immune cells’ infiltration in the tumour immune microenvironment (TIME) [7]. Local TIME could also have a systemic impact on ICIs’ response in a metastatic situation [8,9]. Therefore, researchers have investigated the possibility of increasing TIME infiltration by immune cells to induce a response to immunotherapy in ICI-insensitive tumours [7,10]. Given the local and systemic immunological responses brought on by radiation, combining ICIs with radiotherapy (RT) may overcome response resistance [10]. Tumour response outside the RT field demonstrates the immunogenic cell death mediated by RT (known as the “abscopal effect”) [11,12,13]. Nevertheless, this effect is uncommon and might be influenced by TIME characteristics [14,15]. In this review, we review variations in the metastatic TIME according to the host organ and its impact on ICI outcomes. We also discuss the consequences of RT-induced modifications on the TIME and the systemic immune system to guide future development of immunotherapy-RT combinations in currently non-responding tumours.

## 2. Metastatic Location Matters

The immune microenvironment varies between organs and is influenced by multiple factors, including exposure to external antigens, organ function, and anatomy. Thus, the TIME of metastases differs depending on hosting organs. This variation may account for the different responses to ICIs observed in patients with a similar primary tumour in different metastatic locations. Of the most frequent metastatic locations, the presence of secondary lesions in the liver appears to have the most detrimental impact on the prognosis of patients receiving ICIs.

### 2.1. Liver Metastases

The liver is exposed to many antigens and microbial products that pass through the portal system. These molecules are removed from the blood system by liver cells to prevent an inflammatory reaction [16,17,18]. A wide variety of immune cells exist in the liver sinusoids, including Kuppfer cells, which produce caspase-3 mediated signals to induce T cells’ apoptosis (such as Fas-ligand [FasL] and TNF-receptor apoptosis-inducing ligand). T cell function is further inhibited by immunosuppressive molecules, including PD-L1, IL-10, and transforming growth factor (TGF)-β, which are secreted by liver-specific antigen-presenting cells (APCs, such as Kuppfer cells, hepatic stellate cells, and sinusoidal endothelial cells). In addition, sinusoidal endothelial cells and natural killer (NK) cells stimulate immunosuppressive regulatory T cells (Tregs), which inhibit the action of T cells. However, this state of tolerance favours the establishment and growth of metastases in the liver [16,17,18,19]. Kuppfer cells, neutrophils, and sinusoidal endothelial cells also contribute to the creation of the liver pre-metastatic niche by secreting adhesion molecules that are recognised by cancer cell receptors.

Once established, liver metastases (among frequent metastatic locations, the liver metastases) induce immune modifications that lead to a “systemic immunosuppressive state”. This phenomenon has been inferred from preclinical models that indicate that LM resection decreases systemic immunosuppression [18]. Studies in mice have revealed a reinforcement of FasL-mediated T cell apoptosis by CD11b+ myeloid-derived suppressor cells (MDSCs), which were particularly enriched in the LM TIME [20]. In addition, mice with LM had increased activation of Tregs (CTLA-4, PD-1, and high inducible co-stimulatory molecule [ICOS]), which is known to contribute to immunosuppression (see Figure 1) [21]. A clinical consequence of this immunosuppressive state is poor prognosis for patients with LM, which has been observed in several cancers regardless of systematic treatment [8,9,22,23,24,25,26,27,28,29].

Particularly regarding ICIs, LM-mediated immunosuppression has a significant impact on oncological outcomes [8,20,29,30,31,32,33,34,35,36]. A retrospective study by Tumeh and colleagues included two cohorts of 223 and 113 metastatic melanoma patients, respectively, and one cohort of 165 metastatic NSCLC patients treated with pembrolizumab monotherapy [8]. Compared to lung and brain metastases, LM were found to be an independent poor prognostic factor significantly associated with a shorter progression-free survival (PFS) in all three cohorts. The lower density of CD8+ T cells observed in LM TIME compared to that observed in non-liver metastases may account for this difference. Moreover, non-liver metastases from patients with synchronous LM had a lower CD8+ cells density than those from patients with no LM [8]. In a study by Yu and colleagues, LM were also associated with a reduced objective response rate (ORR) and a reduced overall survival (OS) in melanoma (*n* = 182) and NSCLC (*n* = 279) patients treated with ICIs, compared to those of patients with brain or lung metastases. Interestingly, this association was not observed for targeted therapy in melanoma patients or chemotherapy in NSCLC patients [20]. A retrospective report by Osorio and colleagues evaluated metastatic NSCLC (*n* = 214) and dMMR carcinoma (*n* = 78) patients treated with PD-1 inhibitors and found that the liver was the most frequent site of progression in both cohorts (NSCLC: 24.0%; dMMR carcinoma: 29.7%). Although the response to PD-1 blockade was uniform among synchronous metastases, progression was found to be heterogeneous. The authors hypothesised that the immunotolerance of the TIME in LM explained why the liver was the first progression site following PD-1 blockade therapy [36].

The negative impact of LM has been observed even in cancers with a poor intrinsic response to ICIs. Wang and colleagues conducted a study of 95 MSS CRC patients treated with PD-(L)1 immunotherapy. Patients without LM had a superior ORR (19.5%) and PFS (4.0 months) compared to those with LM, for whom no objective response or increased PFS (1.5 months) were demonstrated [9,36]. Similarly, a reduced response rate was observed in patients with LM who received anti-PD-1 and anti-CTLA-4 combination therapy. In a retrospective cohort of 140 metastatic melanoma patients who received first-line treatment with dual ICIs, the best outcomes (ORR and PFS) were found in patients with lung metastases, whereas comparatively poor results (ORR, PFS, and OS) were observed in patients with LM [37].

### 2.2. Non-Liver Metastases

The lungs, like the liver, are a common metastatic site because they are also exposed to a variety of foreign molecules (e.g., tobacco, traffic pollution, inhaled microorganisms). Thus, tolerance mechanisms are necessary to prevent inappropriate immune reactions. In the lungs, such mechanisms include epithelial cells, airway macrophages, mucociliary action, and commensal microbioma [38]. These mechanisms have the potential to favour a pre-metastatic niche formation, often due to overexpression of the cytokine CCL2, which is derived from the primary tumour and leads resident macrophages and endothelial cells to produce endogenous Toll-like receptor 4 ligands such as S100A8 and SAA3. These effectors facilitate the recruitment of MDSCs and increase vascular permeability, supporting tumoral cell implantation [39]. In addition, γδT cells account for up to 20% of T cells in the lung ecosystem and may promote tumour growth by inducing neoangiogenesis and evasion by restricting the action of APCs, recruiting MDSCs, and secreting immunosuppressive molecules such as IL-4, IL-10, and TGF-β (see Figure 2) [40].

Bone is another common metastatic site in various solid malignancies, particularly prostate cancer, NSCLC, renal cell carcinoma, and breast cancer [41]. The bone environment is considered immune tolerant due to the abundance of Tregs and MDSCs and the lack of NK and cytotoxic T cells [42,43,44]. Primary tumour cells can generate systemic molecular signals that interact with osteoblasts and osteoclasts to promote angiogenesis, cancer cell adhesion, and bone resorption. The combination of these actions generates a pre-metastatic bone niche (see Figure 3) [43,45].

For a long time, the brain was considered an immune-depleted organ due to the presence of immunosuppressive molecules and low T cell infiltration in patients with primary brain tumours [46,47,48]. However, lymphatic vessels have been recently found in regular brain sinuses, transporting immune cells and molecules to the cervical lymph nodes, the first lymphatic relay [47]. A large population of immune cells has also been found in the dura and meninges around the brain, coming through channels directly from the skull bone marrow [49]. Additionally, the microglia, a specialised population of resident macrophages in the brain, is considered the key responder to all types of brain aggressions. It phagocytoses cellular wastes, migrates to the site of disease, secretes pro-inflammatory factors such as IL-1β and tumour-necrosis factor (TNF)-α, and has a direct cytotoxic effect [48,50]. However, prolonged microglial activation induces chronic inflammation and destruction of normal brain cells (astrocytes and neurons), which creates a local environment that favours metastatic invasion (see Figure 4) [48,50,51]. Although patients with brain metastases are usually excluded from pivotal ICIs trials, retrospective data from NSCLC, renal cell carcinoma, and melanoma patients suggest that brain metastases are not a prognostic factor for poor ORR or OS [52,53,54,55].

Studies that have examined ICIs in the metastatic setting do not report lung, bone, or brain metastases as an independent poor prognostic factor for oncological response, in contrast to LM [29,30,31,32,33,34,35,36]. One hypothesis for this discrepancy is that T cells, the main effectors of ICI activity, play a less significant role in normal tissue homeostasis, pre-metastatic niche formation, and metastatic growth in these organs compared to their role in the liver. Therefore, their preservation in non-liver organs would support the maintenance of ICI efficacy, as evidenced by higher ORR in patients with lymph node (LN) metastases [31,32,36]. Indeed, LNs are the sites in which the antitumoral response takes place. LNs are continuously stimulated by tumoral neoantigens, which are processed by dendritic cells to induce CD4+ and CD8+ antitumoral responses favourable to ICI treatment [56,57,58]. However, this situation creates a state of local chronic inflammation that favours the recruitment of immunosuppressive cells, such as MDSCs, tumour-associated macrophages (TAMs), and immature dendritic cells. In turn, this environment is favourable for the implantation and proliferation of cancer cells. This condition is further exacerbated by immunosuppressive signals that are released from the primary tumour, particularly IL-10 and TGF-β (see Figure 5) [57,59,60].

### 2.3. Heterogeneity

In addition to metastatic location, TIME heterogeneity is another factor that may affect treatment outcomes, especially in the era of ICIs [20,61,62,63,64,65]. In several malignancies, including NSCLC, melanoma, renal cell carcinoma, and CRC, there is widespread acceptance of the existence of a gene-expression heterogeneity between primary tumours and their metastases, which may impact treatment results. [61,66,67,68]. In many cancers, the TIME in metastases is more immunosuppressive than that of the primary tumours [67,69,70,71,72,73,74,75,76,77,78].

Metastases located in the same organ share similar TIME characteristics, regardless of the primary tumour. García-Mulero and colleagues reported the gene signature of 374 metastases located in four organs (lung, liver, brain, and bone) of various primary tumours. The TIME of lung metastases was the most pro-immunogenic, with a high score for effector cells and a low score for suppressor cells, followed by bone, brain, and liver metastases [79]. The organ affected by metastases may not account for all sources of TIME heterogeneity, but it may partly explain some TIME variations. A study by Angelova and colleagues assessed spatial and temporal TIME heterogeneities in two metastatic CRC patients with a long disease course, due to the availability of multiple tumour samples. Within the same patient, the TIME differed between metastases and between different areas from the same lesion. Additionally, the TIME may vary considerably between metastases that have a parent-child relationship, based on their similar mutational profiles. Globally, the rate of pro-immunogenic mutations seems to decrease in late-onset lesions, which favours the development of immune escape mechanisms and promotes disease progression. However, high immunoscore values, which reflect a high immune infiltration, can also be found in tumour samples collected in later onset of metastatic diseases [80]. Similar TIME heterogeneity has also been reported for metastases arising from the same organ in melanoma and ovarian cancer [81,82].

TIME-based classification systems have been created to account for the effect of TIME heterogeneity on patient prognosis. These systems may be more predictive of patient outcome than the AJCC/UICC TNM classification, as is the “immunoscore” in CRC, for instance [83,84,85,86].

## 3. Immune Priming Effect of Radiotherapy

Dying irradiated tumour cells release tumoral neoantigens and damage-associated molecular patterns (DAMPs), which enhance dendritic cell maturation. In tumour-draining LNs, activated dendritic cells present these neoantigens to naive T cells, which leads to the activation and clonal expansion of antitumoral T cells, which are then released into the systemic circulation. These activated T cells migrate and infiltrate the microenvironment of lesions distant from the irradiated site, where they can induce a tumoral immunogenic cell death [12,13,87]. This description is a simplified model of the RT-mediated immunological response, which involves additional immune cells—such as Tregs, MDSCs, TAMs, NK cells, and dendritic cells—as well as endothelial cells and stromal cells. The “abscopal effect” (i.e., the immunological response outside the RT field) was first described in 1953, and interest in this effect has increased with the emergence of ICIs treatment because the abscopal effect may improve the systemic efficacy of ICIs in insensitive tumours [11,88]. Encouraging preclinical results have been recorded, but a deeper understanding of the RT-induced immunological response is needed to increase the frequency of occurrence of the abscopal effect, which remains scarce in humans [14,89,90].

### 3.1. Local RT Effect

RT can locally modify the TIME, with variations dependent on the RT dose (see Figure 6). Globally, regardless of dose level, increases have been observed in local infiltration by CD8+ T and NK cells [91,92,93,94,95]. With low-dose RT (2 Gy per fraction), the TAMs polarisation changes to a more antitumoral M1 phenotype, whereas with higher dose of RT (>8 Gy per fraction), the induced TAMs M2 polarisation promotes tumour progression through angiogenesis stimulation [96,97,98,99]. The beneficial effects of RT on dendritic cells also vary by dose level. At moderate (8–12 Gy per fraction) and high (>12 Gy per fraction) doses, RT increases the expression of MHC-II molecules and other co-stimulatory receptors for T cell priming. At low doses, it increases the production of pro-immunogenic cytokines such as IL-2, IL-12, and IFN-γ, which promote T cell proliferation [92,100,101]. High-dose RT reduces the number of infiltrating MDSCs, whereas moderate- and low-dose RT enhance this number, creating an immunosuppressive environment [87,102,103]. Tregs are considered radioresistant compared to other immune cells. In a murine model of prostate cancer, the proportion of Treg infiltration in TIME increases linearly from 2 Gy to 20 Gy [104]. However, another preclinical study found that low-dose RT reversed the Treg tumoral infiltration mediated by earlier high-dose RT [99]. Finally, the effects of RT on the vasculature can also impact the metastatic TIME. Immune cell migration is facilitated by low-dose RT because the treatment improves blood vessel permeability [96,105,106]. On the other hand, high-dose RT may lead to major blood vessel damage and restrict tumoral infiltration by immune cells [76,106].

### 3.2. Systemic Effect

As for local responses, RT-mediated systemic immune reactions may vary with RT dose. In a preclinical study, Vanpouille-Box and colleagues discovered that high-dose RT (20 Gy per fraction) activates Trex1 exonucleases, which degrade the cytosolic DNA required for interferon-β-mediated CD8+ cell activation. This activation limits the abscopal response compared to that of moderate-dose RT (8 Gy per fraction) [107]. According to the findings of another preclinical study by Dewan and colleagues, fractionation is another component that may elicit the abscopal effect [107,108]. Mice that received three doses of 8 Gy had a stronger induction of interferon-β than mice that received a single dose of 8 Gy. However, low-dose RT (<2 Gy per fraction) did not induce sufficient direct tumoral cell death to release neoantigens and DAMPs and further increase CD8+ T cell infiltration in distant tumours (see Figure 6), compared to higher doses (>8 Gy) [109].

It is well established that RT can induce lymphopenia that can be long lasting and associated with poor clinical outcomes in several malignancies, whether used alone or in combination with chemotherapy or hormonal therapy [110,111,112,113]. The severity of lymphopenia depends on the RT site and size of the RT fields. Absolute lymphocyte count (ALC) drop may be predicted by the RT dose delivered to haematopoietic organs, such as bone marrow, the spleen, or circulatory system [114,115,116]. In stereotactic ablative RT (SABR), the RT fields are reduced compared to conventional RT, which may limit the severity of lymphopenia. RT-mediated modifications of circulating T cell populations also appear to be linked to the immune environment of the organ hosting the target lesion. In a Phase I study of 35 patients with various primary tumours, SABR was given to patients with lung metastases or LM in combination with ipilimumab. The patients treated with lung SABR had a significantly lower proportion of circulating CD8+ cells expressing ICOS, glucocorticoid-induced TNFR-related protein (GITR), and lymphocyte-activation gene 3 (LAG-3) than patients treated with liver SABR. In addition, 4-1BB, GITR, T cell immunoglobulin, mucin containing protein-3 (TIM-3), and PD-1 were less expressed on circulating Tregs in patients treated with lung SABR compared to patients treated with liver SABR. Some of these markers are immunosuppressive (LAG-3, TIM-3, and PD-1), while others are pro-immunogenic (ICOS, GITR, and 4-1BB). These data do not support the theory that liver RT induces a more immunosuppressive or pro-immunogenic response than lung RT. However, the researchers suggest that this difference reflects an earlier activation of the immune system when the liver is irradiated than when the lung is irradiated [117].

### 3.3. Implications for RT–ICIs Combination

The selection of the site to irradiate may play an important role in promoting the abscopal response. It may also induce objective responses in patients who do not respond initially to ICIs, such as those with LM. Limited clinical data have been reported regarding the impact of the irradiated organ on the TIME modifications and the abscopal effect occurrence. Only indirect indications can be extracted from immunoradiation trials. However, in most cases, these studies evaluated a small number of patients, and no strong evidence can be drawn (see Table 1) [90,118,119,120,121,122,123,124,125,126,127].

LM irradiation can reshape the TIME and reverse systemic LM-induced immunosuppression. In mice, liver RT has been found to increase CD8+ T cells and reduce MDSC infiltration in the TIME. The addition of anti-PD-L1 ICI amplifies this effect by enhancing specific populations of antitumoral CD8+ cells that express Ki67, IFN-γ, and granzyme B. Compared to the use of RT or anti-PD-L1 alone, the combination of these therapies increases CD8+ T cell infiltration in non-irradiated tumours and reverses the systemic ICI resistance induced by LM [20]. However, data regarding TIME modifications induced by liver RT in humans remains limited. A combination of different RT dose levels with ICIs could also be effective at reversing the immunosuppressive state of LM, as was demonstrated in a preliminary post hoc analysis of a Phase II study. This study evaluated the combination of high- and low-dose RT with continuing ICIs beyond progression in NSCLC and melanoma patients. The lesion-specific response rate after low-dose RT was higher for LM (71%, *n* = 7) than for lung metastases (29%, *n* = 17). In addition, T- and NK cell infiltration were enhanced in lesions treated with low-dose RT [128,129].

Irradiation of tumour-draining LNs is recommended as part of curative treatment for several cancers. However, since the immune response takes place in the tumour-draining LNs, RT in these areas may be detrimental when using ICIs [130,131]. The abscopal effect was reduced among mice that were irradiated at the tumoral site and the tumour-draining LNs compared to those irradiated at the tumoral site only. The irradiation of tumour-draining LNs decreases the stem-like CD8+ T cells within the irradiated tumour, in distant tumour sites, and in their respective draining LNs [130]. This observation occurred regardless of whether RT was combined with ICIs. OS was lower in mice irradiated at the tumour site and tumour-draining LNs compared to those only irradiated at the primary tumour, whether ICIs were directed against PD-1 or CTLA-4 [132]. Moreover, avoiding large volumes and haematopoietic organ irradiation enables reduction of RT-induced lymphopenia, which is associated with a lower OS in patients treated with an RT-ICI combination [133]. This finding suggests that a preserved systemic immune system is required to maximise the results of the RT-ICI combination. Thus, stereotactic treatments that reduce the total irradiated volume should be preferable in immunoradiation trials that aim to increase the occurrence of abscopal effect.

There is growing agreement that multi-site RT, rather than single-site RT, should be used in combination with ICIs [134,135,136,137]. In a preclinical pancreatic tumour model, the response in a non-irradiated tumour was stronger and faster in mice with two irradiated sites than in those with only one [138]. Due to the molecular heterogeneity across metastases, targeting several lesions enhances the variety of tumoral neoantigens generated. The processing of these neoantigens by dendritic cells increases the T cells’ repertoire, which may result in a less stereotypical antitumoral response [135,136]. However, when the metastatic burden is important, SABR may not be safely delivered to all sites without jeopardising the protection of the surrounding healthy tissue. In this case, a combination of moderate-dose (6–12 Gy) or high-dose (>12 Gy) per fraction RT on a single target and low-dose RT (total dose of ≤10 Gy in fractions ≤2 Gy) on all other tumour sites could be an option to promote tumoral infiltration by cytotoxic T cells and subsequently induce significant regression in lesions receiving low-dose RT. This phenomenon, called the “RadScopal effect”, has been observed in several preclinical trials [135,139,140]. Patel and colleagues proposed a decision-making process to select the site and dose of RT based on the patient tumour burden and immune function. They suggested giving an ablative treatment with high-dose SABR to all sites in oligometastatic patients to improve both local control and the immune priming effect. Combining ablative RT on only one lesion with low-dose RT on every other lesion (or even the entire organ) in patients with high-volume disease may effectively stimulate the immune system without causing major toxicity. However, for immunocompromised patients, they expected only a poor immune reaction to be induced by low-dose RT. Therefore, they recommend radical doses for these patients [135]. In addition, having ALC before RT could potentially guide the radiation oncologist to making such a decision, since high ALC is predictive of abscopal response and PFS in RT-ICI combination [135,141].

## 4. Conclusions

Since the development of ICIs, the characterisation of the TIME has become an essential part of metastatic patient care. The TIME can vary depending on the metastatic location and is shaped by the involved organ. Further, the metastatic location impacts the systemic anti-tumour immune effects since patients with liver metastases exhibit a poor response to ICIs compared with patients with metastases in other organs. Combination with RT is currently being investigated to improve ICI response, given the immune priming effect induced by radiation. RT-mediated immune reaction is not only local but also systemic and can lead to responses in non-irradiated tumour sites in rare cases, which is known as the abscopal effect. Future RT-ICI combinations must consider several technical aspects of the RT, such as dose, fractionation, and RT site, to increase the occurrence of such distant responses. These aspects require further investigation to guide radiation oncologists in optimising RT-ICI combinations. Multi-site RT should also be considered, if safe, to increase the diversity of the anti-tumour response.

## Figures and Tables

**Figure 1 biomolecules-13-00689-f001:**
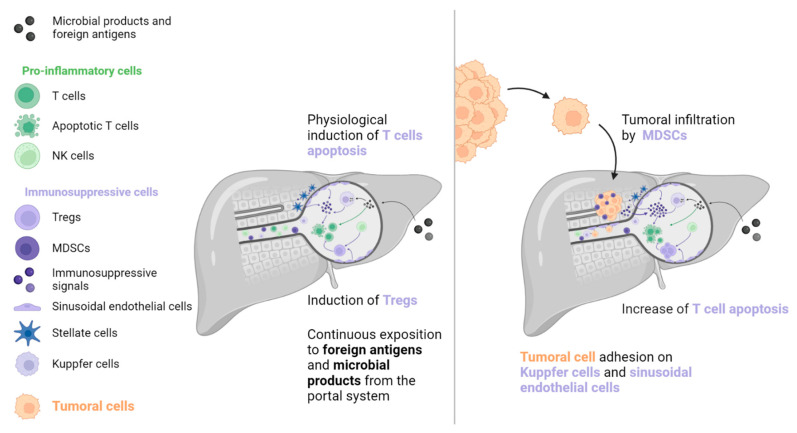
To prevent immune system overactivation, the liver has tolerogenic mechanisms that are primarily mediated by the induction of T cell apoptosis. When a tumour is present, this pathway is strengthened by molecular signals originating from tumoral and TIME cells (such as MDSCs, Kuppfer cells, sinusoidal endothelial cells, and stellate cells). T cell inhibition and depletion may explain the low efficacy of immune checkpoint inhibitors in patients with liver metastases. MDSC: Myeloid-derived suppressive cell, NK: Natural killer, TIME: Tumour immune microenvironment, Treg: Regulator T cell.

**Figure 2 biomolecules-13-00689-f002:**
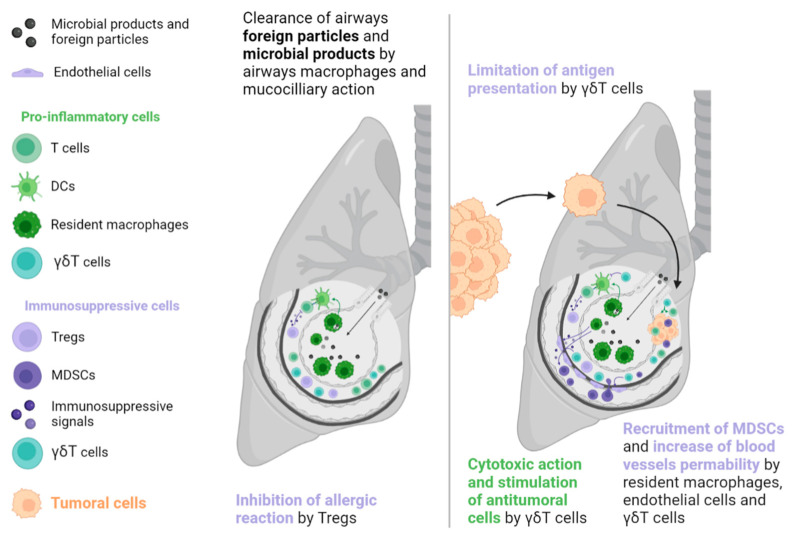
Tolerogenic mechanisms in the lungs involve less T cell apoptosis than those in the liver. In the presence of a tumour, a particular lymphocyte population called γδT cells can have ambiguous effects on tumour development. These cells exert a direct cytotoxic action and an indirect antitumoral action via the stimulation of T cells. However, they can also facilitate tumour development through the secretion of immunosuppressive signals, recruitment of immunosuppressive cells, promotion of neoangiogenesis, and limitation of antigen presentation. DC: Dendritic cell, MDSC: Myeloid-derived suppressive cell, Treg: Regulator T cell.

**Figure 3 biomolecules-13-00689-f003:**
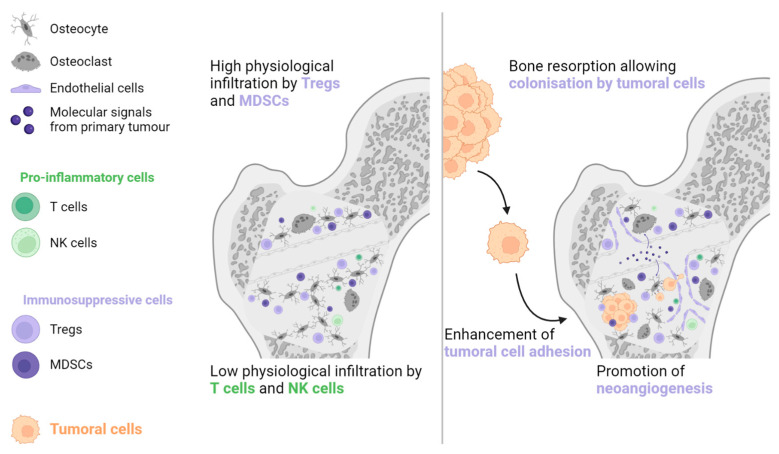
The physiological abundance of Tregs and MDSCs, combined with the paucity of antitumoral immune cells, is the main reason why bones are frequently invaded by metastatic lesions. The primary tumour generates molecular signals, which contribute to the formation of a pre-metastatic niche by increasing bone resorption, stimulating neoangiogenesis, and enhancing tumour cell adhesion. MDSC: Myeloid-derived suppressive cell, NK: Natural killer, Treg: Regulator T cell.

**Figure 4 biomolecules-13-00689-f004:**
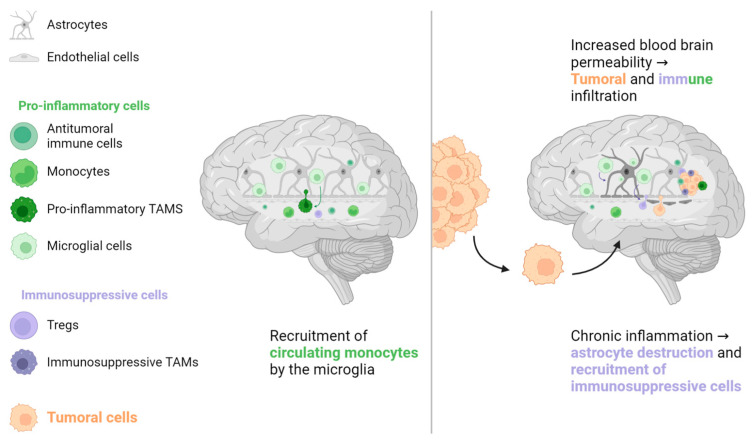
The brain has its own population of macrophages, the microglia. These cells, the main responders to brain injury or aggression, stimulate the recruitment of immune cells, mainly monocytes, from systemic circulation. In the case of chronic inflammation, microglial cells can directly damage normal brain cells such as astrocytes, which are crucial for preserving the blood-brain barrier. Increased blood-brain barrier permeability can promote tumour cell migration and recruitment of immune cells, some of which have an immunosuppressive phenotype that favours tumour cell implantation. TAM: Tumour-associated macrophage, Treg: Regulator T cell.

**Figure 5 biomolecules-13-00689-f005:**
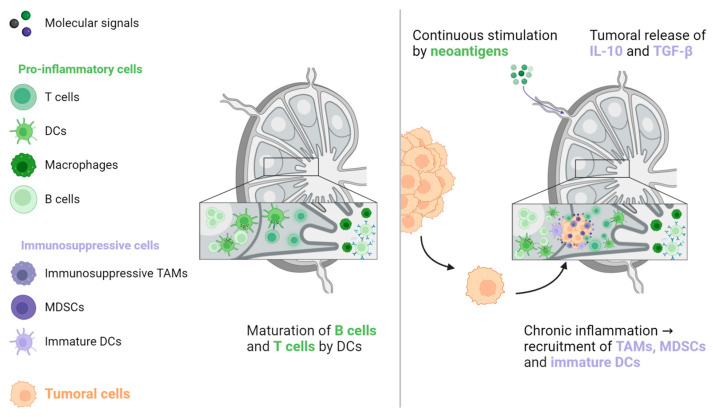
The activation of naive B and T lymphocytes takes place in lymph nodes. When these lymphocytes are continuously triggered by neoantigens from the primary tumour, antitumoral immunity is counterbalanced by immunosuppressive cells, including TAMs, MDSCs, and immature DCs. In addition, the primary tumour generates immunosuppressive signals that promote tumoral cell implantation and proliferation. DC: Dendritic cell, MDSC: Myeloid-derived suppressive cell, TAM: Tumour-associated macrophage, TGF-β: Transforming growth factor-β.

**Figure 6 biomolecules-13-00689-f006:**
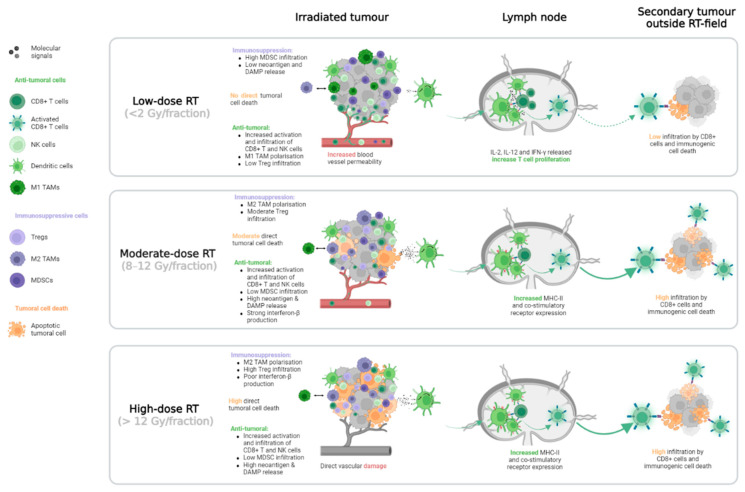
Summary of modifications to the tumour immune microenvironment and systemic reactions induced by different RT dose levels. Unlike moderate- or high-dose RT, low-dose RT does not induce enough direct cell death to sufficiently release neoantigens and DAMPs, resulting in a weak distant immunological response. In the case of high-dose RT, induction of the Trex1 exonuclease decreases interferon-β production, which is required for a strong abscopal response. Compared to high-dose RT, low-dose RT may reduce Treg infiltration in the irradiated tumour and promote M1-polarisation of TAMs, thereby inducing an antitumoral response. The effect of RT on MDSCs is also dose dependent: moderate and high-dose RT decreases the infiltration of these immunosuppressive cells, whereas low-dose RT increases infiltration. Dendritic cells play a key role in CD8+ T cell priming in tumour-draining lymph nodes. At low-dose RT, this phenomenon is mostly due to dendritic cells’ production of IL-2, IL-12, and IFN-γ, but at higher doses, it occurs due to dendritic cells’ enhanced expression of the MHC-II receptor. Finally, the reshaping of the tumour immune microenvironment is also dependent on the effect of RT on blood vessels. Low-dose RT increases vasculature permeability, which allows immune cells to migrate to the tumoral site. In the case of high-dose RT, immune cell migration is restricted due to direct RT-induced vasculature damage. DAMP: damage-associated molecular pattern, MDSC: Myeloid-derived suppressive cell, MHC: Major histocompatibility complex, NK: Natural killer, RT: Radiotherapy, TAM: Tumour-associated macrophage, Treg: Regulator T cell.

**Table 1 biomolecules-13-00689-t001:** Summary of prospective trials evaluating abscopal response in patients treated with RT-ICI combinations. Only studies reporting abscopal response occurrence according to the RT site were included. Abscopal response was defined as the occurrence of a complete or partial response as the best response observed in metastases outside the RT field, according to response evaluation criteria in solid tumours (RECIST) v1.1. GM-CSF: Granulocyte-macrophage colony-stimulating factor, ICI: Immune checkpoint inhibitor, LN: Lymph node, NSCLC: Non-small cell lung cancer, ORR: Objective response rate, PD-1: Programmed cell death, RT: Radiotherapy, TNBC: Triple-negative breast cancer.

Author	Primary	ICI	*N* RT Fractions	Dose per RT Fraction (Gy)	Total Population		Irradiation of LN Metastasis		Irradiation of Lung Metastasis		Irradiation of Liver Metastasis	
					*N*	Abscopal Response (%)	*N*	Abscopal Response (%)	*N*	Abscopal Response (%)	*N*	Abscopal Response (%)
Sundahl, 2019 [90]	Melanoma	Nivolumab	3	8	20	9 (45.0)	7	5 (71.4)	5	1 (20.0)	1	0 (0)
Twyman-Saint Victor, 2015 [118]	Melanoma	Ipilimumab	2 or 3	6 or 8	20	4 (20.0)	2	0 (0)	9	1 (11.1)	3	1 (33.3)
Hiniker, 2016 [119]	Melanoma	Ipilimumab	1–15	2.5–25	20	6 (30.0)	3	0 (0)	6	3 (50.0)	2	0 (0)
Sundahl, 2018 [120]	Melanoma	Ipilimumab	3	8–12	13	1 (7.7)	3	0 (0)	3	0 (0)	1	0 (0)
Maity, 2021 [121]	Melanoma	Ipilimumab	2–3	6–8	22	5 (22.7)	6	1 (16.7)	10	2 (20.0)	3	1 (33.3)
Formenti, 2018 [122]	NSCLC	Ipilimumab	3	6 or 9.5	21	7 (33.3)	1	0 (0)	13	5 (38.5)	3	0 (0)
Qin, 2020 [123]	NSCLC	Atezolizumab	2–5	6–10	12	3 (25.0)	4	2 (50.0)	2	1 (50.0)	1	0 (0)
Sundahl, 2019 [124]	Bladder	Pembrolizumab	3	8	18	4 (22.2)	8	2 (25.0)	8	2 (25.0)	-	
Voorwek, 2019 [125]	TNBC	Nivolumab	3	8	12	1 (8.3)	5	1 (20.0)	1	0 (0)	1	0 (0)
Welsh, 2019 [126]	Various	Ipilimumab	4 or 10	6 or 12.5	95	9 (9.5)	-		52	6 (11.5)	43	3 (7.0)
Kong, 2022 [127]	Various	PD-1 inhibitor & GM-CSF	3	5 or 8	48	9 (18.6)	16	3 (18.6)	7	3 (42.9)	7	2 (28.6)

## Data Availability

No new data were created.

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
