# Peer review of "Implications of the Organ-Specific Immune Environment for Immune Priming Effect of Radiotherapy in Metastatic Setting"

_biomolecules, 2023, doi:10.3390/biom13040689_

Round 1

Reviewer 1 Report

Thank you for the opportunity to review this study, which aimed to review comprehensively the published evidence on the TIME of different location of metastasis, the effects of radiotherapy on TIME, and the potential advantages of combining radiation with ICIs therapy for metastatic setting, which is a topic with rising importance. My comments/questions for the authors are as follows:

1. The manuscript needs extensive revision for language and grammar.

e.g. In line 33 the description, “…such has programmed cell death (PD-1)…” should be “as”.

In line 40 the description, “…such has prostate cancer…” should be “as”.

In line 46 the description, “…a response to immunotherapy in tumours ICI unsensitive” should be “insensitive”.

In line 194 the description, “…or immatures dendritic cells,…” should be “immature”.

In line 391 the description, ”The TIME can vary depending onto metastases…”should be “on”.

2. The author did not mention what the abbreviation “LM” is for. Please revise it.

3. The use of terminology in immunology in this manuscript is inappropriate.

e.g. In line 29 the description, “…the major impact of immune-infiltrating cells” I suggest revise the term “immune-infiltrating cells” to “tumor-infiltrating immune cells”.

In line 225 the description, “Decreasing immunoedition pattern over time could…”I suggest revise the term ” immunoedition”.

4. In line 61-62 the description, In that respect, the liver seems to present with the biggest impact on the metastatic response to ICI compared to other organs.” The statement was unclear. Please revise it.

5. In line 224-226 the description, “Furthermore, TIME composition varied considerably between parent and child metastases. Decreasing immunoedition pattern over time could explain acquisition of escape mechanisms to immune system and disease progression.” The statement was unclear. Please revise it.

6. In line 311-3 the description,” Lung patients had significantly lower proportion of circulating CD8+ cells expressing ICOS, glucocorticoid-induced TNFR-related protein (GITR) and lymphocyte-activation gene 3 (LAG-3) than liver patients.”. The use of the phrases ”lung patients” and “liver patients” is not appropriate. Please re-write the sentence.

7. One phase II trial could be provided for your reference about the positive effect of liver radiotherapy combined with immunotherapy on metastatic NSCLC and melanoma patients. Preliminary post-hoc analysis of this trial (NCT02710253) showed a lesion-specific response rate after low-dose RT that was higher for liver metastases (71%, n=7) compared to lung metastases (29% in lung, n=17). Besides, T- and NK cell infiltration was enhanced in lesions treated with low-dose RT.

**Patel RR, He K, Barsoumian HB, Chang JY, Tang C, Verma V, et al. High-Dose Irradiation in Combination With non-Ablative Low-Dose Radiation to Treat Metastatic Disease After Progression on Immunotherapy: Results of a Phase II Trial. Radiother Oncol (2021) 162:60–7.

Reviewer 2 Report

Pierrard et al reviewed the immune microenvironment in different metastatic organs such as liver, lung, bone and brain, with a focus on the discussion of the effect of radiotherapy and immune checkpoint inhibitors. Overall, it is a comprehensive review with logical thinking and decent discussion. And both figures and table make manuscript easier be understood. I only have two tiny suggestions regarding the current version:

1. Assuming LM means liver metastases. Authors need to indicate it when first mentioning it.

2. Line 221, format issue needs to be fixed.

Reviewer 3 Report

In this manuscript, authors discussed the tumor immune microenvironment (TIME) composition at different metastatic sites, including, liver, lung, brain, bone and lymph node.  Authors further discussed the local and systematic effects TIME composition by radiation therapy (RT), provide considerations of dose, the fractionation, and the site of irradiation combined with immune therapy may improve patients’ outcome.

Overall, this is a very thoughtful and informative review manuscript.

Minor points:

1.      Line 221-222 wrong paragraph break.

2.      Line 400, need to start with capital letter.

Round 2

Reviewer 1 Report

Thank you for the revise and detailed response letter. I have no further question.